# Countering the Tragedy of the Health Care Commons by Exnovation: Bringing Unexpected Problems and Solutions into View

Willemine Willems 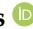

Athena Institute, Vrije Universiteit, De Boelelaan 1085, 1081 HV Amsterdam, The Netherlands; w.l.willems@vu.nl

**Abstract:** In the health sciences and policy, it is common to view rising health care costs as a tragedy of the commons, i.e., a situation in which the unhampered use of a resource by rational individuals leads to its depletion. By monitoring a set of outcomes, not only the costs but also patient experience and population health, simultaneously, it is claimed that the "triple aim" approach changes what is rational for health care stakeholders and, thus, can counter the rapidly rising health care costs. This approach has an important limitation: it reduces the monitored innovations to merely their outcomes; yet, how health care professionals and patients give shape to care delivery remains invisible. To get a more in-depth understanding of the consequences of adopting such an approach, in this article I use the method of exnovation instead. Exnovation foregrounds the everyday accomplishments of health care practices to enable reflection and learning. I draw on an ethnographic study into an innovation in care delivery aimed at rendering it more sustainable: Primary Care Plus. I reflected with both professionals and patients on what happened during 40 Primary Care Plus consultations. By presenting and analyzing three of these consultations, I foreground what is rendered invisible with the triple aim: improvisations, surprises and habits unfolding in practice. With exnovation, health care innovations can provide fertile soil for creating new forms of sustainable care that can help prevent the impending exhaustion of health care systems.

**Keywords:** sustainable care; exnovation; tragedy of the commons; reflection; health care innovation; care delivery

## 1. Introduction

Rising health care expenditures in European countries and in the US have raised grave concerns about the future of their health care systems [1–4]. For some, this has been a reason to typify health care as a "common", and the growth of costs as the cause of a "tragedy of the commons" [5–7]. A tragedy of the commons is a type of economic situation in which the self-interested and unhampered use of openly accessible resources by individuals leads to its depletion. Often used to explain the problem of overfishing, or the depletion of fossil fuels, at its core it is a problem of rationality [7]. For individuals, it is not rational to abstain from using the commons as much as they can, because they get the full benefit of its use, while the damage is limited because it is shared with the collective populace. In such a context, whether it is the use of fossil fuels, fishing, or receiving health care, there is one designated solution: "As in all tragedies of the commons, the great task in policy is not to claim that stakeholders are acting irrationally, but rather to change what is rational for them" [8] (p. 761).

With the influential triple aim approach, Berwick et al. developed a framework for changing what is rational for health care stakeholders [8,9]. Health care managers, policymakers, and researchers are encouraged to strive for a collective instead of an individual goal, called the "pursuit of triple aim" [8] (p. 760). Stakeholders are stimulated to not pursue one goal individually but instead three interdependent collective ones

simultaneously (costs, patient experience, population health, and, in later literature, a fourth aim: providers' satisfaction [10]).

In the past decade, the triple aim approach gained traction in health policy and health sciences research worldwide [5,9,11–17]. The approach has been adopted on the level of health systems in countries such as the US and Canada, but also by a wide variety of regional health care initiatives in countries as diverse as England, New Zealand, and Singapore [9]. A significant insight after 7 years of experience with this approach, noted by Whittington et al., is the importance of establishing a learning system [9]. According to the authors, a learning system is a comprehensive assessment method consisting mainly of outcome measures, which allows for comparing performance with the preset aims of an initiative, enabling the sustaining of improvements and learning within the lifespan of the intervention [9].

Using the triple aim approach, organizations commit to an iterative implementation process, directed by the outcome measures of the learning system. Pursuing the triple aim is thus a balancing act in which an organizational intervention is shaped gradually by the assessment of its performance, in an effort to reach a positive outcome in terms of the aggregate of its costs, patient experience, and population health [8,9]. In the operationalization of the framework, the intervention is typically conceived as a test [9], that is, an experiment, a controlled trial. The hypothesis is then that the intervention under examination improves health care according to the three aims (i.e., the predefined criteria). The outcomes of an intervention group, consisting of patients who have received new care, are systematically compared with the outcomes of a non-intervention group consisting of patients with similar problems who received care in a normal setting. Hence, with such an approach, the causal relationship between the intervention and improvement in terms of the triple aim can be established or rejected [14].

By foregrounding outcome measures, the learning system of the triple aim approach helps provide a clear overview, and, as such, a means to compare its outcomes with its aims and with other health care interventions. Yet, such an approach comes with an important shortcoming: it reduces the reality of innovation to its outcomes, rendering the innovation itself its daily reality, and the processes taking place during the intervention become invisible. This is especially true when using questionnaires, which collect explicit and conscious knowledge from the respondents, while most people have a "practical blindness" for the actions they undertake regularly in mundane contexts, such as providing or receiving health care [18] (p. 18).

Until now, scholarly work on the triple aim approach has been mainly concerned with the operationalization and application of the approach in different types of health care contexts. For example, several articles reflect on how the triple aim approach could inform the reform of specific health care practices [19,20], or they use the framework to assess the performance of existing systems, innovations or practices [21–23]. Other articles aim to gather evidence for the impacts of the approach [24], or to assess whether the approach is suitable for application in specific contexts [15,16]. More theoretically or methodologically oriented scholarly work focuses on the refinement or extension of the three dimensions of population health, quality of care, and costs [9,10,21,25,26]. Even though some of the articles in this body of literature raise points of critique on the framework itself and/or how it is used in practice, none have critically reflected on the learning system's emphasis on outcome measures and their consequences.

In this article, I adopt the method of exnovation in order to explore and critically reflect on what is rendered invisible when using the outcome-oriented triple aim approach. Like the triple aim approach, exnovation is a method that enables learning, but instead of focusing on outcomes for the purpose of learning, it foregrounds and reflects on the daily work processes of delivering and receiving health care [27–29].

More specifically, the article draws on an ethnographic case study of a health care innovation called Primary Care Plus that was set up, assessed and gradually shaped by the outcome measures of patient experiences, population health and costs (triple aim) [12–14].

Primary Care Plus is a new form of specialist health care delivery intended to address non-urgent medical problems that can be solved with one or two consultations. Primary Care Plus is aimed at reducing costs, while safeguarding accessibility and quality of care. By exploring the observations and reflections of three Primary Care Plus consultations in detail, I aim to bring into view that which is rendered invisible with the learning system of the triple aim approach, contributing to a broad perspective on sustainability in health care.

## 2. Materials and Methods

### 2.1. Study Design: Primary Care plus as a Case Study of Innovative Health Care Delivery

The article draws on the observations and interview data gathered as part of an ethnographic case study into Primary Care Plus, conducted at two Primary Care Plus centers. As a new form of care delivery, Primary Care Plus is a combination of elements of both primary and secondary care. In the Dutch health care system, primary care is provided by medical professionals, such as general practitioners and physiotherapists, whose consultations are directly accessible for patients and are paid for by obligatory health care insurance. Secondary care denotes non-urgent care provided by medical specialists in a hospital or a private clinic. Patients can only access such care with a referral letter from a general practitioner. Patients pay for part of this care themselves (deductible); the obligatory health insurance pays the rest. As a combination of primary and secondary care, Primary Care Plus is delivered by a medical specialist (secondary care) in a general practice or a neighborhood health care center (primary care). Patients need a referral from their general practitioner (secondary care), but the consultations are fully paid for by the health care insurance (primary care). In contrast to specialist care delivered in the outpatient clinic of the hospital, in the Primary Care Plus center, the specialist merely examines patients via an anamnesis and a simple, low-tech physical examination. Without available high-tech diagnostics, specialists are expected to rely mainly on their medical knowledge and experience.

The intervention was expected to improve care in terms of costs, patient satisfaction and health. Relocating specialist care from the large, impersonal, and bureaucratic hospital to a small-scale neighborhood center would improve patient experience. Care would be more accessible, not only in terms of travel time but also in a transitive sense; patients were expected to feel more at ease. The cost reduction gained was expected not only because of the low overhead cost in comparison to outpatient clinics in the hospital but also because the low-tech environment of the centers would discourage expensive treatment decisions that are redundant from a medical point of view. Population health was expected not to be affected by the relocation because if patients needed more extensive specialist care, they could be referred to the hospital.

Primary Care Plus is an informative example of innovation in health care, aimed at making care more sustainable. In the promotion of intervention, the rising health care costs were presented as a tragedy of the commons, with an urgent call for making health care more sustainable. It was designed and assessed, following the principles of the triple aim approach. Ethnographically following the daily practice of Primary Care Plus is thus an especially suitable case for exploring what the triple aim approach renders invisible.

### 2.2. Methodology: Exnovation

For the purposes of this study, I have adopted the methodology of exnovation, developed to direct the researchers' gaze to the "mundane, to the implicit local routines, to what is already in place" [29] (p. 76). Exnovating Primary Care Plus consultations thus means foregrounding what is done there and then, in contrast to the triple aim approach, which foregrounds the *outcomes* of what was done there and then. Note that the term "exnovation" is currently being used in a variety of ways in scientific research. In this study, "exnovation" does not denote the process of "addressing ( . . . ) what needs to be done to fully establish the new" [30] (p. 991), nor does it denote the process by which "practitioners turn away

from an existing practice" [31] (p. 29), but instead a research process by which the skills and ingenuity of professionals and their mundane routines are foregrounded [29]. The common denominator of these ways of using the term is the intended contrast with the term "innovation". While other studies emphasize that transition processes not only entail introducing new elements but also leaving behind old ones, in this study, "exnovation" denotes the opposite: it aims to improve practices not by shedding old elements but by "paying attention to what is already in place" [32] (p. 3).

As a method, exnovation was developed within the broad family of practice theories that emerged as a result of the practice turn in social sciences [33–40]. In line with this turn, I track what patients and professionals *do* during the consultation. To be more specific, I use Pickering's approach, one that foregrounds what scientists do in practice instead of viewing science as a formal procedure for establishing representative knowledge of the wider world [41]. Pickering thus presents science as "a mangle of practice", an "evolving field of human and material agencies reciprocally engaged in a play of resistance and accommodation, in which the former seeks to capture the latter" [42] (p. 23), which places its unpredictability at center stage [41]. Viewing the Primary Care Plus consultations as a mangling of practice means foregrounding what professionals, patients, the furniture, the building, and the absence of high-end technologies "do" when navigating the playing field of resistance and accommodation, that is, when they deal with the tensions and moments of harmony that arise in interactions with each other and with the material context in which these interactions take place.

### 2.3. Data Collection

The data collection (Table 1) for this study consisted of two parts. First, I conducted semi-structured qualitative interviews with medical specialists who regularly work at the center and in the outpatient clinic of the hospital (8 in total), and with regional general practitioners who had the option to refer patients to the Primary Care Plus centers (10 in total). While the specialist interviews took 55 min on average, the interviews with general practitioners lasted on average 30 min. The purpose of the interviews was to gain an in-depth understanding of why they participated in the project, how they experienced their involvement, and what they considered to be fruitful or problematic in both the care concept and in the way it has evolved in practice and over time. This part of the study functioned as a preparation for the exnovation methodology.

**Table 1.** Data collection process.

| Phase | Method | Interviewee/Location | Number | Duration |
|---|---|---|---|---|
| Phase 1: preparation for exnovation | Qualitative semi-structured interviews | General Practitioners | 10 | 30 min (average) |
| | | Primary Care Plus specialists | 8 | 55 min (average) |
| Phase 2: exnovation | Ethnographic observations followed by reflective interviews | 3 General Practices | 37 consultations | 36 h (total) |
| | | 2 Primary Care Plus Centers | 40 consultations of 14 different medical specialists | 75 h (total) |
| | | 4 Outpatient clinics | 27 consultations | 20 h (total) |

The purpose of the second part of the data collection was to make the mundane work of the specialist and patients in the consultation room explicit (exnovation). For this purpose, I combined ethnographic observations with reflective interviews. In 2016, I attended consultations at two Primary Care Plus centers, three regional general practices and at four outpatient clinics of the local hospital in 2016. In total, I spent 75 h at the centers, 36 h at the general practices and 20 h at the outpatient clinics, observing interactions between doctors and patients during the consultations and in the waiting rooms, and holding short informal conversations with doctors, patients, receptionists and assistants. In this period of ethnographic observation, in all, I followed 14 medical specialists, attending a total of 40 of their consultations in the centers and 27 at their outpatient clinics, and

met with three general practitioners, attending 37 consultations to make ethnographic observations. During the consultations, I made notes about what was done and what was said, simultaneously identifying those moments that would be informative for reflection afterward. Before and after each consultation, I held short reflective interviews with the patient and doctor participating in the consultation separately. This allowed me to not only observe doctors and patients accomplishing care activities but also to understand why and how these activities unfolded. During the interviews, I made use of my position as a non-medical outsider, unaware of and unable to know how such things are usually done in medical practices. In this way, it also became necessary for doctors to explain the habitual behaviors that they take for granted because they are widely accepted among medical professionals.

The notes made during the consultations were based on in situ observations and made with pen on paper; all short and longer interviews were audio-recorded. Within three days after each day of attending consultations at a Primary Care Plus center or a general practice, I transferred the notes to a Microsoft Word document, adding insights and observations that had not been noted yet. I transcribed the interviews verbatim.

### 2.4. Data Analysis

The data analysis roughly consisted of three steps. The first step was a first round of "coding", which was carried out inductively, with the purpose of identifying recurring themes. I read the transcripts in detail and marked moments, routines, and reflections of interest. One of my supervisors read the same transcripts, in order to see if the same moments and reflections stood out to both of us. We discussed what the marked excerpts showed and how they were informative for the research. The identified themes of interest gave a direction for a literature search, on the basis of which I constructed the analytical lens described in the Section 2.5, which I used for the third step of the analysis.

The second step of the analysis was a selection of three consultations, of which the observations and reflective interview transcripts provided the most evident insight into the identified themes. In a more general sense, these three consultations gave insight into processes that would stay out of view with the triple aim approach. As such, the study is a form of falsification, or what in social sciences is called critical reflexivity. Presenting only three consultations might not give scientific validation for predictive purposes or for establishing causal relations, but they represent scientifically valid sources of information for exploring the complexities of care delivery that stay out of view with the triple aim approach [43] and, as such, form a strong basis for formulating points of critique.

The selected consultations are presented in this article with "thick description", a way of describing an observed situation that is used in ethnography/social anthropology, one that goes beyond factually noting events (in detail); it is a first interpretation step in which the meanings of gestures, actions and words for the observed persons themselves are also presented [44,45]. Such thick description was possible because the period of ethnography was extensive, allowing me to get to know the medical professionals and the organization because I not only attended the consultations but also had long and short conversations before and after the consultations with the observed participants.

In the third step, I analyzed the thick descriptions with the analytical lens developed on the basis of the first step. This analysis is presented in the Results section.

### 2.5. Analytical Lens

The systematic foregrounding of what people (and objects) do in practice, conducted during the data collection and first two steps of the data analysis, brought sets of activities into view that proved to be similar to what Strauss [46,47] coined as "articulation work". Articulation work entails the often-unnoticed types of activity involved in connecting the different parts of organizational processes required to make them run fluently. Note that Strauss does not use the common, dictionary meaning of the verb "to articulate", that is "to pronounce something clearly and distinctly" or "to express something fluently", but the

less common one of "to consist of segments united by joints" [48]. For example, Primary Care Plus care is not ready to be delivered whenever the consultation rooms are reserved, and the responsibilities are distributed among professionals. In terms of articulation work, the separate parts of the workflow need to be "integrated" and "meshed" [49]. Such work can be achieved by formal planning and scheduling, but it also "requires implicit and intangible efforts, such as the bringing together of social worlds" [50] (p. 65) (also refer to [49,51]). Such articulation work has a temporal aspect (planning), but also a spatial dimension, called "mobility work", i.e., "the moving about of people and things as part of accomplishing tasks" [52] (p. 131).

Analyzing the data, it became clear that, in the case of the care work accomplished in Primary Care Plus consultations, another dimension of articulation work is crucial: aligning the differing understandings of what good care entails. For patients and doctors alike, it is crucial not only to find and communicate the right diagnosis and treatment decision but also to convince the other or to be convinced that the result is the right outcome of a fair process. This not only depends merely on social and medical norms but also on moral convictions of what good care entails. Finding a common ground during the consultation is thus crucial for the durability of the decision; patients who are not convinced that the diagnosis or treatment decision is the right one, or that the way they were treated by the doctor is fair, will very likely find another doctor to consult and possibly another one after that. In order to accomplish durable decisions, it is required that doctors and patients "unite" the relevant moral "segments by joints". In analyzing the observations and interviews, I thus focused on explications of actions involved in accomplishing such durable care.

To bring the moral articulation work into view, I highlight the differing understandings of good care by further substantiating the concept, using Boltanski and Thévenot's sociology of situated judgment [53] as an analytical lens. In their ethnographic study of work settings in which people achieve decisions together, despite their differences and without resorting to violence [53,54], Boltanski and Thévenot focus on what they call operations of justification and critique. Such operations, usually verbal, but possibly also embedded in actions [55], consist of placing the particular situation to which the decision in question pertains according to a moral order of worth, i.e., in "coherent vocabularies of argumentation and justification that are each organized around one vision of the common good" [56] (p. 4). Boltanski and Thévenot present a non-exhaustive typology of six such orders of worth: the "market" (economic growth), "industrial" (efficiency), "fame" (reputation), "domestic" (emotional proximity and trust), "inspired" (inspiration) and "civic" (autonomy and equality) [53]. In the ethnographic observations, three of these six orders were found: the civic, industrial, and domestic orders. An example of the adoption of the civic order was a patient explaining that she was satisfied with her received care because of the doctor treating her as equal to any other patient, irrespective of the impression of being difficult that she has given in the past. In contrast, a patient trusting a doctor because he spoke in the local dialect fits the domestic order of worth, because emotional proximity is emphasized. Specialists justified their decisions by emphasizing the importance of care trajectories running efficiently (industrial), or by referring to previous encounters in which they had formed a more personal understanding of the patient (domestic).

In my analysis of the data, I thus identified those actions involved in accomplishing durable care and the ways in which patients and doctors retell those actions in terms related to equality, emotional proximity or efficiency as operations of justification. These actions, and how they are justified, represent the moral dimension of articulation work. Adopting the concept of justification *work* resonates with other literature in which a type of work is foregrounded that is commonly taken for granted [57,58]. To be more specific, I make visible how patients and doctors successfully attune their differing understandings of what good care entails in the consultation room.

### 2.6. Ethical Considerations

Preceding my attendance at consultations and in the waiting rooms, I approached medical specialists who worked at the Primary Care Plus Centers and the general practitioners who would regularly refer patients there. With them, I arranged under which conditions I could attend their consultations and speak with their patients. I announced my presence to patients via posters in the waiting room that showed a photograph of myself and a short description of the research. At the care locations, I approached patients in the waiting room, gave them an information sheet about the research, and asked whether they were willing to participate. If they were, I first explained in detail how I was planning on storing and using the data; second, I would ask them to fill in and sign an informed consent form. The research design was approved by the Medical Ethical Commission of the Academic Hospital in Maastricht (METC/azM-UM).

### 3. Results

In this section, I present observations and quotations from the ethnographic fieldwork; more specifically, of the three selected consultations. Exploring these with the analytical lens developed above, I reflect on how doctors and patients accomplish the goals of Primary Care Plus together, and, as such, give insight into what is rendered invisible with the triple aim approach. The tree diagrams give an overview of how the observations and quotes relate (Figures 1–3).

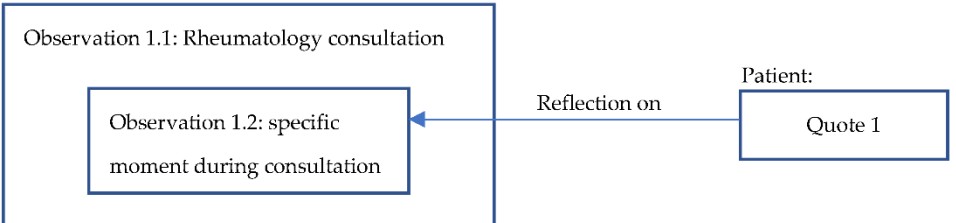

**Figure 1.** Reflection and observation overview Section 3.1.

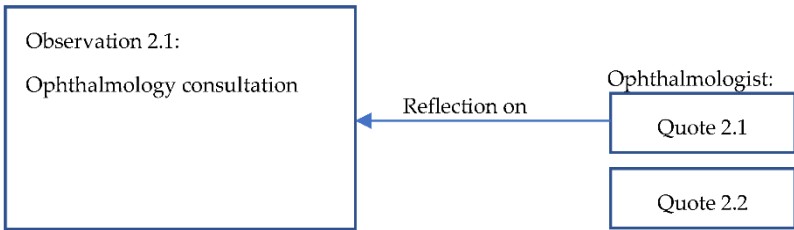

**Figure 2.** Reflection and observation overview Section 3.2.

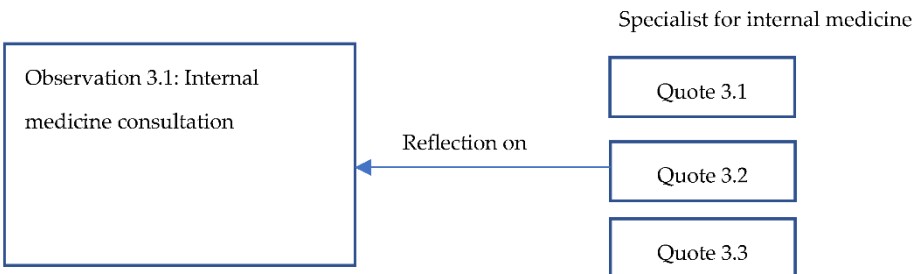

**Figure 3.** Reflection and observation overview Section 3.3.

### 3.1. Crafting Durable Decisions

In the first observed consultation, the Primary Care Plus rheumatologist examines a woman in her mid-20s. She has been suffering from painful joints for years. The general

practitioner had ordered some tests and X-rays approximately two and four years ago. The images did not show anything out of the ordinary. The general practitioner did not see any reason to order more tests or to refer the patient to a specialist. Afterward, the patient felt that her problem was not taken seriously because nothing was done to find out what is causing her pain. That is why she pushed her general practitioner to refer her to a specialist. Preceding the consultation, she confided that she was rather worried, especially because her mother recently told her about a 26-year-old colleague who had recently been informed that she will probably not live much longer than six months because of a serious medical condition. (see Figure 1).

Observation 1.1.: The consultation starts with an anamnesis, during which the rheumatologist asks questions about the patient's pain at the specific moment of the consultation, what she has felt over the past years, and whether she has any family members suffering from rheumatoid arthritis. In terms of the physical examination, the rheumatologist tests the patient's joints by moving her arms, legs and upper body and by pushing on certain spots. Then, they assess the images and test results from a few years earlier, together.

Near the end of the consultation, the rheumatologist says: "The most important thing is that there is no sign of rheumatoid arthritis. Your elbow . . . ", she hesitates, " . . . I don't know what to make of it, but I think it is more likely a problem of the muscles than of the joints. It is probably caused by a bit of hypermobility syndrome." Meanwhile, touching the painful spot on the patient's arm, she says, "But if that would be the case, I don't understand why it hurts here." Looking at the patient's posture she says: "If you learn to use your body differently, your joints have to carry less weight." Then she switches to the patient's painful back: "Work on your posture. Stand in front of a mirror and check whether you are standing up straight."

In the final minutes of the consultation, the rheumatologist suggests optional next steps without making these seem urgent: an MRI (magnetic resonance imaging) to see the state of the patient's tendons, an ultrasound scan to examine the elbow. "But if you want that," she says, "you would need to go to the outpatient clinic [at the hospital]. We can also start with some anti-inflammatory pills and wait and see how that goes." Patient: "Soon we will have a break from volleyball training over the summer. Maybe it is a good idea to see how my elbow feels after?" Rheumatologist: "That is a good idea. In the advice letter I will tell your general practitioner to check if the elbow is still hurting after the summer. If it is, he can always prescribe anti-inflammatory medicines."

At first sight, the interactions described in observation 1.1. are a typical example of a doctor gathering the information required to rule out the serious condition of RA, subsequently sharing her insights with the patient and making a shared decision about the next steps. Yet, viewing the observed consultation as a mangle of practice brings other aspects to the foreground. Interactions between doctor and patient can then be revealed as a process of navigating the human agencies of the consultation like a playing field. More specifically, using the lens of justification work, the discontent felt by the patient toward her general practitioner and his way of handling her symptoms becomes of central concern. At the Primary Care Plus Center, the patient leaves the consultation room satisfied, even though the specialist had not been able to give her an unambiguous answer as to what is causing her pain. In the interview directly after the consultation, the patient explains what made her feel content:

> Quote 1: The conversation [with the specialist] was very pleasant. I was happy she checked everything, and also, that she wrote it down and filed it, in the computer. ( . . . ) Her examinations were more extensive [than those of the general practitioner] ( . . . ) and it made a difference that I have seen it myself now. ( . . . ) Not like: "Oh, that looks good", click, click [mimicking her general practitioner quickly clicking through the X-rays]. ( . . . ) She [the rheumatologist] said: "When it looks smooth, there is nothing wrong". And then, I saw for myself that it looked smooth, which is why I feel confident about it.

In Quote 1, the patient refers to a moment at the beginning of the consultation room when the rheumatologist and patient together viewed the older test results. These moments are more fully described in observation 1.2:

Observation 1.2: After the physical examination, the patient and the specialist sit down at the desk again. Specialist: "Your general practitioner has ordered tests for inflammation markers two and four years ago. It all looked good. Let's see if I can find the X-rays." She turns the computer screen towards the patient.

While describing out loud what she sees, she constantly points to different parts of the image on the screen. "We are looking now at the shape of the hips. Do the different parts fit well? As you can see the joints are beautiful and smooth. When we conduct these kinds of examinations, we closely look, for example, at whether the edges of the joints are frayed. On this picture we do not see that at all."

The X-ray image of a hand appears on the screen. "This is the thumb, these are the finger bones, ( . . . ). Here as well, I check whether everything is nice and smooth. As you see, there are no scores or little pieces of bone missing." She points at a specific part of the hand and says: "Here, it looks very nice as well!"

Next is an X-ray image of the patient's upper body. "The vertebras of your spine are beautifully aligned. The distances between each of them is similar. If you would change your posture, which is somewhat hollow, you could possibly take away quite a bit of pain." The rheumatologist gets up from her chair to show her how to correct a bad posture: "Bend the knees lightly, tuck under your pelvis", she says while illustrating it by doing it herself. Then: "None of the diagnostics that we have right now show abnormalities."

Viewed through the lens of justification work, observation 1.2 brings a crucial part of the mundane work into view: that rheumatologist and patient work together to align orders of worth. The patient leaving the room satisfied, crucial for accomplishing a durable decision, is not only the result of the quality of the information she received but also largely reflects the effectiveness of the process of justification work. In her explanation of why she is content—in Boltanski and Thévenot's terms, this is an operation of justification [43]—she places the specific moment in which the rheumatologist explains the images in a civic order of worth. Comparing this moment to her recent unpleasant care experiences at the general practice, she emphasizes that, in this case, "she could see it for herself", bringing the values of autonomy and equality in the decision-making process to center stage. She does this, however, not in terms of having sufficient knowledge, or of getting as many possible tests as possible, but in a reflection on how the process unfolded. A doctor "click-clicking" through X-rays makes her feel shunted off and dismissed, yet the Primary Care Plus rheumatologist explained to her in detail what was visible on the images and how to deduce a decision from that. This enabled the patient to become a decision-maker herself, side by side with the rheumatologist.

From the above, I draw three conclusions. The first one concerns the order of worth emerging in the justification work. In the process unfolding in the consultation room, the civic order of worth emerged as the appropriate order for the patient in this specific situation. Afterward, the patient deemed the consultation to be an example of good care, crucial for the durability of the decision, because the common good of patient autonomy was nurtured. That this is the appropriate order of worth for her at that moment was not clear to her before she entered the waiting room; she grew to see it as crucial during the consultation and in her reflection afterward. A second, related point is that the processes of justification work are dynamic and creative. For the patient, autonomy is not something that she possessed beforehand and that needs to be safeguarded; it is an accomplishment; it is created through the interactions in the consultation room. By sharing her interpretation of the older test results and her routine of making a diagnosis, the rheumatologist enabled the patient to become an autonomous decision-maker in the context of the consultation. Third, such justification work is not merely the work of the rheumatologist in the consultation room during the consultation. It also entails the process by which the patient anticipates what is coming next, her experience of the interactions, and her reflections afterward.

With a process that extends before and goes beyond the timespan of the consultation, the patient navigates the agencies around her, figuring out for herself, step by step, what good care entails in her specific situation. The civic order of worth emerged during the consultation, but not solely because of the rheumatologist's explanation; among other events, the mangling of her previous experiences and the presence of a researcher asking questions cocreated the conditions for it.

As such, I suggest that justification work is not merely a fleeting moment of interaction in a limited time frame, but that, instead, it has more extensive potential for building the capacities of both patient and doctor to make durable decisions, and, thus, contribute to sustainable care. Crafting a durable decision is, thus, a creative process in which both doctor and patient are involved in feeling out and tuning in to each other's orders of worth. In this case, the accomplishment of sustainable care cannot be defined by a set of criteria beforehand; it is not a matter of changing what is rational for patients and doctors using the health care commons, it is a creative process of reflection and alignment.

### 3.2. Unforeseen Problems

The second consultation that I discuss in this section is an ophthalmology appointment at the Primary Care Plus Center. The patient consulting the doctor decided to have her eyes checked when her optometrist could not identify the right prescription glasses. (see Figure 2).

Observation 2.1: The consultation follows the same course as most ophthalmology consultations at the center. After the patient sits down on the examination bench, the doctor starts a conversation about her symptoms, followed by a vision test and a slit lamp examination. At the end of the consultation, the eye doctor explains what he found and what he thinks should be done about it. Eye doctor: "There are two issues. You have a bit of cataract. I don't think you need to do anything about that right now. Also, your retina is getting thinner and there are little spots on it." He continues to ask some more standard questions, and then: "Do you have tiles in your bathroom?" The patient seems surprised while answering the question affirmatively. Eye-doctor: "Check now and then if the lines are still going straight when you look at them. When they start meandering, then you should immediately come to the hospital. That means the small blood vessels in your eyes are starting to grow rampantly. But for now, we should not jump to conclusions. ( . . . ) Just come back if your sight is getting worse."

Observation 2.1 can be viewed as a particularly literal example of how the main rationale of Primary Care Plus's sustainability plays out in practice. As explained in the Materials and Methods section, holding consultations in a low-tech neighborhood center is expected to encourage specialists and patients to refrain from taking expensive treatment decisions that are redundant from a medical point of view. At the Primary Care Plus Center, the ophthalmologist only has two diagnostic tests at his disposal: a vision test and a slit lamp examination. Because an ophthalmologist is dependent for observations on technical devices, this is an especially limited set. In the brand-new eye clinic at the hospital where he works on the other days of the week, he regularly uses around 20 directly available apparatuses for a whole range of routine diagnostics. The new eye clinic is set up as a systematically designed workflow space that guides patients through an efficient procedure, one in which the diagnostics associated with their condition can be conducted in a single day, without much waiting time. In the above Primary Care Plus consultation, he finds signs of an early stage of macular degeneration. While not harmful at such an early stage, in later stages the medical condition can lead to blindness. In the high-tech eye clinic, it is a routine decision to perform an optical coherence tomography (OCT) to determine a baseline value, in case of any signs of macular degeneration. Yet, at the Primary Care Plus center, where an OCT could still be ordered by referring the patient to the outpatient clinic, the eye doctor makes the alternative, low-tech treatment decision of asking the patient to keep an eye on the lines between the tiles instead.

In Section 3.1, I showcased justification work in which the doctor navigated human agencies; observation 2.1 foregrounds a doctor navigating *material* agency. To be more specific, the decision-making of the eye doctor revolves around the material agency of bathroom tiles and absent high-tech apparatuses. Within the promotion and presentation of Primary Care Plus, using low- instead of high-tech diagnostics is mainly justified by its sustainability, i.e., by emphasizing how it safeguards the quality and accessibility of health care by increasing its cost-efficiency. Such a justification is thus framed in a civic-industrial order of worth because it places the common goods of equal access and efficiency at center stage. Yet, in practice, such types of justification do not always feel natural. After the consultation observed in the above excerpt, the patient felt reassured, but it left the eye doctor frustrated. In the short interview, he confided that, while in principle, he was a supporter of the intervention, taking such a treatment decision in a consultation feels unsatisfactory; keeping an eye on the lines between the bathroom tiles is not an *optimal* alternative to an OCT, it is merely *medically acceptable*. He brings up a similar situation to explain the complexity of navigating absent apparatuses:

> Quote 2.1: If we look at this patient; in the hospital it is a standard procedure to make an OCT. ( . . . ) Such an OCT is a small procedure for the patient. It is not burdensome at all! It takes only a minute to get the needed information! ( . . . ) Imagine you see a patient of around 80 years old [at the Primary Care Plus center]; he has a bit of vision decline, though still seeing 70%. On the retina some shifts in pigment are visible. ( . . . ) When I see the same patient in the center, I think, 'well, he is still seeing 70%, he is managing. Should I send this patient who is a bit older, who might have come with special transportation, who might be dependent on his rollator to the hospital while he has already made such an effort to come here?

Quote 2.1, in which he reflects on the situation described in observation 2.1, gives insight into how the justification work in observation 2.1 unfolded. Crucial in the quotation is the doctor's operation of critique, which consists of inserting an imaginary patient into the argument. Invoking the imagined situation, now with a dependent elderly man, he appeals to feelings of emotional proximity. Within the invoked domestic order of worth, the absence of diagnostic apparatuses acquires a different meaning, because the values that become of central concern are different from the ones in the civic-industrial order that is dominant in marketing material. In the evoked order of worth, emotional proximity is the highest common good, and, thus, the obligation of the doctor to care for his patient because he is dependent becomes the appropriate frame of action. The decision to order an OCT for an elderly patient acquires a different meaning as well. In the high-tech eye clinic, it is a routine action, part of the optimized workflow of the clinic, which involves neither time nor effort; in the center, it is still the optimal medical option but also a burdensome procedure and a complicated journey, one that might be hard to organize for a dependent 80-year-old. The justification work conducted by the doctor in this case is not a matter of feeling out and tuning in to what the patient deems good care but, thus, revolves around good care within the material constraints of the decision.

The relocation of care resulted in a shift in orders of worth and values of concern but, relatedly, it also caused what I call moral discomfort. Reflecting on this type of Primary Care Plus dilemma makes the ophthalmologist swear, sigh, and bring up other annoying aspects of the relocation:

> Quote 2.2: Why is it the specialist who must make the effort to come to the location [where care is delivered]? Sometimes I arrive gasping for breath, because something in the hospital has taken more time than expected or I had to do an emergency ultrasound. ( . . . ) The time it takes to travel here, I could have used to see patients.

While meant as a harmless intervention that would gently push doctors and patients to make more sustainable decisions, the arrangement also has a detrimental effect on the

doctor's daily work satisfaction. In some of the consultations, he follows the values he deems important in the situation but because of the sub-optimal treatment decisions that he ends up taking, he starts questioning the setup of the intervention. His moral discomfort is a weak version of "moral distress", i.e., "knowing what to do in an ethical situation, but not being allowed to do that" [49] As a professional, he knows what the optimal decision for this medical condition would be, but when navigating the material agency of the Primary Care Plus consultation, he ends up deciding otherwise.

From the above, I draw four conclusions. First, in the consultation room, the doctor and patient not only navigate human but also material agency, in this case, the "resistance" that the *absence* of new, high-tech apparatuses are giving and the accommodation that the enrolled bathroom tiles are offering. Second, the justification work that the eye doctor conducts in the consultation room extends before and goes beyond the 10 min of the consultation. It has become part of how he creates somewhat coherent narratives about his past and future decision-making. Justification work is, thus, also the effort of aligning his moral past, present and future. Third, since every doctor has different past experiences and convictions about what good care entails, the consequences of relocation of care from the outpatient clinic of the hospital to a neighborhood center are not easy to predict. How such relocation affects the decision-making of a doctor, and whether it will cause moral discomfort, depends on how the doctor understands the new situation in contrast to the one that has become normal to him. For this doctor, the relocation made him place the dilemma in a domestic order of worth; for another doctor, some other order of worth might become important after such a relocation. A fourth conclusion is related to the fact that the specialist only voiced his discontent after repeatedly reflecting on specific instances of decision-making. Foregrounding the complex reality of accomplishing durable care reveals that the relevant and unexpected effects of an intervention are not always visible in the outcomes of an intervention but might only come up when reflecting, again and again, on the process.

### 3.3. Creating New Care Routines

The third consultation I scrutinize in this article is an internal medicine consultation at the Primary Care Plus center. The patient suffers from overactive bowels. Because he has some other medical issues as well, he has become quite familiar with his regular specialist in the outpatient clinic of the hospital. Placing the referral in a domestic order of worth (proximity and trust), he explains his discontent with being referred to a doctor who is not familiar with his many medical problems. For his general practitioner, however, the referral had been the appropriate choice. The civic-industrially justified Primary Care Plus intervention is mainly seen as a support of the general practitioners' gatekeeping function. Bypassing it is permissible if there are clear medical indications that the patient needs specialist care. Within this order of worth, trust and familiarity are not adequate reasons for legitimately deviating from the norm. (see Figure 3).

Observation 3.1: Like most of this specialist's consultations at the center, the appointment starts with an extensive anamnesis, in which the broad range of medical problems of the patient are addressed: overactive bowels, heart palpitations, an IgA deficiency (an inborn disorder of the immune system), anxieties caused by the forthcoming birth of his second child, tensions at work, et cetera. During the conversation, the specialist, again and again, expresses interest with both words and body language, communicating that the non-medical issues are also relevant to her. After the anamnesis, she extensively examines the patient's abdomen and bowel. Observation 3.1 is from the interactions between specialist and patient near the end of the consultation when they make a treatment decision together.

The specialist opens the results of the latest blood tests up on her computer screen. The tests show "no signs of inflammation", "kidney function normal" for the patient's age, and a few "minor deviating values" regarding his "liver cells". "There are only two problems," she concludes, "the IgA deficiency and the deviating liver values. I will advise

your general practitioner to order an ultrasound of your liver. Besides that, I will contact your specialist in the hospital to see if there is anything else we need to do."

The patient is still somewhat dissatisfied: "I would like to get a check-up at the hospital every few months." Specialist: "Let me first talk to the specialist about it. And you need to make an appointment with the general practitioner to see how to proceed from here. "Any other questions?" Patient: "I am still worried." Specialist: "You have coped with these problems your whole life. You have grown up with them. There are only some loose ends. Some things you need to do to prevent problems in the future". "Can those values go up again suddenly?", he asks, still worried. Specialist: "Until now it hasn't bothered you too much. So, what can you do? Just go on with your life. You can turn 100 with these kinds of problems."

Patient: "Is there anything I can do about the sounds from my bowels? I am ashamed when I sit in a quiet space or when I am at a meeting." Specialist: "There is not much you can do. Of course, you could just tell people about your problem. There are a lot of people who have such problems. ( . . . ) Whenever you are gassy, people will think, 'oh right, he has such and such problems'". After some more questions, the patient leaves the consultation room. Clearly relieved, he sighs: "That was absolutely a very pleasant consultation".

The above shows a specialist patiently pushing for a conservative treatment decision while countering the demands of a patient insecure about his own physical condition and the capacities of his general practitioner. The repeated requests for regular appointments with his specialist of internal medicine at the hospital and for more tests seem to be induced by the patient's need for trust. His attempt to gain trust (domestic common good) through numbers (the type of knowledge deemed most reliable in an industrial order of worth) is a trope also identified in research conducted in other contexts [50]. As such, it is a typical example of how compromises between orders of worth emerge. Within the domestic-industrial order of worth, numbers give a sense of control and predictability that would help regain his trust over his situation and the course of his medical conditions. However, his demands are not met. While going home without a clear diagnosis, newly ordered tests, or regular appointments for his bowel condition, he nevertheless goes home satisfied with the consultation.

After observing several of the specialist's Primary Care Plus consultations and extensively reflecting with her on the specifics, it became apparent that her seemingly spontaneous actions are part of a carefully constructed routine for dealing with a broad range of patients' trust issues. These small and more elaborate routine actions are diverse. One example is the length of the consultations. Because the Primary Care Plus centers are new, the consultation hours are not yet as tightly organized as in the hospital. This situation of newness has allowed the specialist to arrange with the assistants to make only one 30-minute consultation per hour, giving space to take more time with each patient. Another example of her routines is that whenever she starts her lengthy anamneses, she sits on her desk chair, facing the patient on the other side of the desk. While typing her notes, she turns her computer screen, enabling the patient to read while she types. Specialist:

> Quote 3.1: When I turn the screen, they say, "Yes, yes, that is exactly what I mean."
> It makes people feel understood ( . . . ). It also gives the patients some structure,
> so they will stay a bit concrete and to the point.

In this way, the anamnesis becomes a process concerned not only with gathering medically relevant information but also with restoring the patient's trust. During the conversation, also attempts to restore trust by giving medical and non-medical advice and by complimenting the patient about his body. "Such beautiful slim ankles!", and "What a flawless and smooth inside of the bowel", she exclaims, while physically examining the specific body parts on the examination bed.

Another important trust mission concerns the patients' relationships with their general practitioners. Even though the Primary Care Plus specialists are allowed to directly refer patients to the outpatient clinic of the hospital, as a rule, she refrains from doing this.

Allowing the general practitioner to stay in charge might help improve her relationship with the patient. Telling the patient that she will ask another specialist (observation 3.1.) for advice is a purposeful part of her mission to restore the patient's trust in his general practitioner, as well:

> Quote 3.2: He will have to deal with his general practitioner for years and years. He needs to trust her to be a serious doctor who has taken him and his problems seriously. Telling him that I need to consult the professor specialized in this subfield of internal medicine is a way of saying, "This is a complicated clinical picture, I can imagine your general practitioner does not know all the ins and outs of it." ( . . . .) I also need the input of the super-specialist.

For the specialist in internal medicine, collaborating in the Primary Care Plus intervention is mostly motivated by her mission to fight what to her is one of the most urgent problems of today's health care: the increase in patients' trust issues. According to her, 40% of the patients who consult her at the outpatient clinic of the hospital do not have a medical problem but a "trust problem". Even though the centers seem poorly equipped for countering such issues—short consultation hours, unfamiliar doctors, a maximum of one follow-up consultation—she believes that the centers can nevertheless help with the endless care trajectories of patients, a cycle that is damaging to patients.

> Quote 3.3: If they come to the hospital, they first see an intern, then the supervisor. In the meantime, several tests have been done that are not necessarily related to the problem the patient came in with. ( . . . ) If you are insecure as a patient, you will think "Oh, I must have something really serious" and become really worried. This is how they enter a cycle in which more worries cause them to demand more tests, which will cause yet again more worries, and so on.

Note that in quote 3.3, the specialist also conveys a connection between trust and tests. This time, the numbers are not seen as the appropriate means to gain trust but as critically contributing to the diminishing trust of patients in their own bodies.

The justification work was effective in the consultation. As the consultation progressed, a relationship of trust between specialist and patient emerged. The patient's initial discontent about being referred to an unfamiliar specialist resolved when his worries were addressed with a care routine specifically developed to help patients like him. Before he leaves the consultation room, the treatment decision achieved is endorsed by the doctor and is sufficiently comfortable for the patient.

From the above, I draw two conclusions. The first one pertains to justification work. These observations are a confirmation of the findings in the previous sections. Justification work is the work conducted by doctors and patients in the consultation room, aimed at aligning moral orders of worth by navigating agencies in a play of resistance and accommodation. Foregrounding the justification of this consultation shows it to be part of a broader effort in which—in this case—the specialist works toward creating a somewhat coherent idea of what good care entails in particular types of situations. Becoming involved in the Primary Care Plus intervention fitted her mission to solve the problems she had identified in her work practice in the outpatient clinic of the hospital. The experimental character of the intervention provided experiences and temporal and conceptual space to iteratively shape a new type of sustainable care: Primary Care Plus care, as a solution to the damaged trust of the patients she sees in her consultations. Such work is not merely medical, it is moral and social and extends beyond the time span and walls of the consultation room. The second conclusion pertains to exnovation as a method for gathering insights that contribute to recognizing and assessing sustainable care. What I learned from the above is that the specific, new form of care developed by the specialist is the result of an unpredictable process that stays out of view or that is not recognized as valuable when an assessment method is focused on outcome measures alone. That is to say, the intricate practices of justification work that unfold when providing health care in new conditions

produce new health care phenomena that can feed a process of finding creative new ways of delivering sustainable care.

## 4. Discussion

In the above, I analyzed in detail three consultations through an analytical lens that enabled foregrounding the justification work conducted in the Primary Care Plus consultations. The aim was to explore and critically reflect on what is made invisible when adopting the triple aim approach; to be more specific, what lessons are missed out on when a learning system consisting of a comprehensive assessment method is adopted to gradually and iteratively give shape to a health care organization. I found that when learning is based merely on outcome measures, unexpected problems and iteratively created but promising new forms of care stay out of view. In the first and third consultations, I traced the practical, medical and moral solutions that specialists adopt to deal with the widespread "trust issues" of patients; the second consultation showed how the relocation of care delivery can cause moral discomfort. That these aspects stay out of view when learning is based on outcomes is problematic, because newly created routines and emerging problems are particularly valuable for improving health care delivery practices.

### 4.1. Policy Implications

That the triple aim approach, and specifically its reliance on outcome measures, has been welcomed so enthusiastically and widely in health care policy and research is not surprising. The approach perfectly fits the growing political demands for scientific proof of the effectiveness of health care innovations, and the related, growing reliance on qualitative measurements. As such, the focus on outcome measures fits the needs of what Latour calls the "centers of calculation", those venues where unique practices are made comparable, which is required for making policy [39] and, as such, have become the way in which health services and practitioners account for the quality of their care (and its cost). In the sense that a researcher using the methodology asks the professionals and patients to account for and reflect on the decisions made, exnovation is a process of accountability as well. While it does not serve the purpose of calculation, in its own way, it is also informative for health care policy.

From the exnovation described in this article, two important policy implications can be identified. First, basing assessment and learning processes mainly on outcome measures may possibly push out of sight the diverse ways in which patients and professionals deal with moral and social differences in the consultation room. As was revealed in the above examples, the justification work required for making care processes run smoothly, that is to say, for achieving durable diagnostic and treatment decisions, is a creative as well as a dynamic process. I have revealed how orders of worth emerge, values come into being, and routines are iteratively established. Justification work is not a fleeting effort conducted within the walls of the consultation room, it starts before the consultation and extends beyond it. As such, it is involved in how individual patients and professionals unite the moral past, present and future understandings of what good care entails, using joints. How individual professionals and patients conduct justification work is diverse; how such work is conducted, for example, which orders emerge and which are irrelevant, depends on personal tendencies and experiences in the past. When learning is merely informed by outcomes, the rich processes by which doctors use their justification work will not be available for reflection and learning. These reflection and learning processes are crucial for moral capacity-building, which in ethics literature is denoted as the process of moral or social scaffolding [59,60], making them especially valuable for improving the skills of medical professionals.

The triple aim approach and research conducted in this field have become particularly influential in the past 14 years. While health care managers and researchers adopt the framework for designing and assessing regional organizational innovations, it has also informed policy at the level of national health care systems. One of the interesting aspects

of the triple aim approach is its emphasis on learning and improving innovations along the way. However, even though learning is at the core of the approach, without paying attention to the processes of health care accomplishments, such learning might not reach its full potential. Adding a process-oriented method, such as exnovation, to the comprehensive assessment methodologies will add the possibility of harvesting new creative ideas and learning from professional skills for dealing with the diversity of patients and medical problems that medical professionals encounter on a daily basis.

A more specific contemporary health care puzzle for which this research might be informative is the relationship between patient satisfaction and symptoms. Instead of researching whether medically redundant testing counters the anxieties of patients with unexplained medical complaints [61], it could also be informative to look at how general practitioners ascertain what value such a test has for the patient in his consultation room, which depends, among other factors, on what the patient considers to be good care. On the basis of the research presented in this article, I argue that such a skill cannot be acquired by becoming acquainted with the results of outcome-based research; instead, this is more likely by exnovation, i.e., the immersion in a process of enacting justification work and reflecting on what happened then and there, afterward. Because the mundane practices of care accomplishments are so diverse, it is more informative to learn how these events are done well, than to know how patients respond to them on average.

The second policy implication pertains to understanding the rising health care costs as a tragedy of the commons, which is becoming increasingly prevalent in policy and scholarly literature [5–7]. By adopting the concept of the tragedy of the commons, the rising health care costs become a policy puzzle regarding how to redirect the rational behavior of health care stakeholders from their own self-interest to the interests of the collective. With the triple aim approach, managers and policymakers attempt to redirect stakeholders' rational behavior by encouraging them to balance the outcomes in terms of costs, patient experience, health and, in later versions, professional satisfaction. On the basis of the findings of this research, however, I suggest that the focus should not merely be on how managers and research can (re)design an innovation in order to achieve a balance between the three- or four-outcome measures, but instead on how patients and doctors balance the common goods of efficiency, trust and equality in the practice of accomplishing durable care. Most importantly, how can medical professionals reflect on these processes and learn from them? The tragedy of the commons then becomes a collective and practical balancing act that professionals and patients alike will be challenged to take on. Reaping the fruits from such a process, however, is only possible when, on the level of regional and national health care, assessment and policies are not only aimed at improving outcomes but also at improving the reflective skills of health care professionals.

### 4.2. Limitations of the Study

Set up as a means to critically reflect on an influential approach to designing, assessing and innovating health care delivery, the small number of consultations analyzed limits the scope of the conclusions and implications that can be drawn from this study. While tracking and analyzing in detail how three consultations at the Primary Care Plus Center unfolded does give a basis for finding possible flaws in the triple aim approach, and making suggestions for improving health care practice and policy, the study does not enable the prediction of, for example, the effects of incorporating exnovation in a learning system.

### 4.3. Future Research

The research, thus, raises more questions than it answers and, as such, provides fruitful starting points for further research. An important follow-up of this research could be focused on how a process-focused methodology can be added to the triple aim's learning system. Another starting point for future research is the notion of justification work. As a lens, it could be of value for analyzing and reflecting on a diversity of contemporary health care issues, such as patient-centered care, or methodological discussions about what

patient experience is, and how it should be collected and used to inform management and policy.

## 5. Conclusions

The tragedy of the commons has become an important concept in policy and scholarly research about rising health care costs. An important approach in this regard is the triple aim approach, a framework that is promised to provide a means to turn health care stakeholders from pursuing their own interests to pursuing a collective one, "the pursuit of the triple aim". At the core of the triple aim approach is the idea that implementing health care innovations is an iterative process that should be directed by a comprehensive assessment and learning system, mainly based on outcome measures. In this article, I have argued and demonstrated that learning merely on the basis of outcome measures is problematic because it possibly takes the problems, skills and processes involved in accomplishing durable healthcare decisions out of view. A qualitative methodology, such as exnovation, could be particularly valuable for making the triple aim's learning system stronger and more comprehensive.

**Funding:** This research was funded by the Academic Collaborative Center on Sustainable Care, which is an initiative of Maastricht University Medical Center+ and Maastricht University.

**Institutional Review Board Statement:** The study was conducted according to the guidelines of the Declaration of Helsinki, and approved by the Institutional Review Board (or Ethics Committee) of Maastricht UMC+ (protocol code METC 14-4-136.2/jh and 14 August 2015).

**Informed Consent Statement:** Informed consent was obtained from all subjects involved in the study. Written informed consent has been obtained from the patient(s) to publish this paper.

**Acknowledgments:** I would like to thank my Ph.D. supervisors, Tsjalling Swierstra, Jessica Mesman and Philip Vergauwen. for their inspiration and feedback while in the process of developing the analytical framework and analysis that I used for the argument of this article.

**Conflicts of Interest:** The authors declare no conflict of interest.

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
