# Peer review of "Countering the Tragedy of the Health Care Commons by Exnovation: Bringing Unexpected Problems and Solutions into View"

_sustainability, doi:10.3390/su132313082_

Round 1
Reviewer 1 Report
The study investigates the tragedy of the health care commons by experimentation.
- Despite the high-quality data and the really hard-work that has been done by the author(s), the study could benefit from better defining research questions, stating the goals more clearly and linking with appropriate theoretical or policy literatures.
- Overall, the quality of writing is very good. Nevertheless, I have to note that introduction and conclusion parts could be improved. The manuscript does not include clearly formulated hypotheses - contribution. While the aim of the paper is clearly submitted, the innovation of the paper is missing. It should highlight novelty of this study.
- It should be also beneficial to use some tables or figures (such as tree-diagrams) as it could make the paper easier to read it. For instance, quotes and observation could be summarized in one or two tables. Thus, each reader could understand the whole process in one quick glance.
- It should be better to avoid using personal pronouns (I instead of we) in the text.
- In addition, conclusion part needs revisions as policy implications are very general while future research, innovation of the study and limitations are missing.
- In general, the present study needs some minor revisions. However, the most important doubt is whether the thematic topic of the manuscript is within the scope of the journal.
Author Response
Dear reviewer,
Thank you very much for reading my manuscript with interest and care. Your comments have been helpful to further improve the text. In the following I will explain one by one what I have done to address the 5 issues that you raise in the report.
First you suggest to rewrite the introduction:
Despite the high-quality data and the really hard-work that has been done by the author(s), the study could benefit from better defining research questions, stating the goals more clearly and linking with appropriate theoretical or policy literatures.
Overall, the quality of writing is very good. Nevertheless, I have to note that introduction and conclusion parts could be improved. The manuscript does not include clearly formulated hypotheses - contribution. While the aim of the paper is clearly submitted, the innovation of the paper is missing. It should highlight novelty of this study.
I gather from these remarks three shortcomings of the text: 1) the article is insufficiently linked to the literature, 2) the novelty of the paper is unclear, 3) hypotheses are missing.
In the new version, 1) I have linked the article more extensively to the large body of Triple Aim literature (lines 118-127, 150-161), 2) this has helped to clarify the contribution of the paper (lines 158-166). 3) I have not formulated a hypothesis, because in the disciplinary field of this paper (medical anthropology) it is not common to do that. For an example seminal work in the field, you could read (Pols, 2006)
It should be also beneficial to use some tables or figures (such as tree-diagrams) as it could make the paper easier to read it. For instance, quotes and observation could be summarized in one or two tables. Thus, each reader could understand the whole process in one quick glance.
This is a great idea. I have added three tree diagrams: (lines 809-816, 949-957 & lines 1124-1134)
It should be better to avoid using personal pronouns (I instead of we) in the text
I know this is unusual in other fields, but I use the first-person singular on purpose, which is common medical anthropology for reasons that have been discussed extensively. An interesting article about this is (Venkatesh, 2013) about the reflective turn in social sciences, which has taken place in the past 30 years. Scholars who use ethnographic methods have been reflecting more and more on their own role in their field work and in the interpretation of the observations. One of the most important conclusions of these reflections is that a field worker and a research subject can never be fully on equal footing. The least that ethnographic researchers can do is to be transparent about their own role in observation and analysis. Using the first-person narrative voice is thus a simple and straightforward first step in being reflective of my own role and position. I hope you understand that it would go against my research norms to change this.
In addition, conclusion part needs revisions as policy implications are very general while future research, innovation of the study and limitations are missing.
I have revised the discussion and conclusion extensively with more specific policy implications (lines 1343-1502), limitations (line 1503-1511) and future research (1512-1520)
In general, the present study needs some minor revisions. However, the most important doubt is whether the thematic topic of the manuscript is within the scope of the journal.
I hope I have addressed the issues sufficiently. Thanks for the valuable feedback!
Pols, J. Washing the Citizen. Washing, cleanliness, and citizenship in mental health care. Culture, Medicine and Psychiatry 2006, 30, 77-104
Venkatesh, S.A. The Reflexive Turn. The Rise of First-Person Ethnography. The Sociological Quarterly 2013 54(1), pp. 3-8.
Reviewer 2 Report
In this manuscript, the author used the method of exnovation to get a more in-depth understanding of the consequences of adopting the Triple Aim approach, which has an important limitation that it reduces the monitored innovations to their outcomes, yet it remains invisible how health care professionals and patients give shape to care delivery. By reflecting with professionals and patients on what was done during 40 Primary Care Plus consultations and analyzing three of these consultations, the author foregrounded that improvisations, surprises and habits unfolding in practice were rendered invisible with the Triple Aim approach. This work is interesting and may provide a fertile soil for creating new forms of sustainable care that can help prevent the impending exhaustion of health care systems. This paper needs to address the following issues in a minor revision.
1. When the author was describing the “Data collection”, it would be better if the author could use a table or a pie chart to demonstrate the process.
2. On page 5, line 254: “This does not only depend only on social and medical norms, but also on moral convictions of what good care entails.” The author should delete the latter "only".
3. The author should double check the format of the references.
Author Response
Dear reviewer,
Thanks a lot for reading the paper and for the comments. Below I one by one respond to the 3 issues you raised in the report.
When the author was describing the “Data collection”, it would be better if the author could use a table or a pie chart to demonstrate the process.
This is a great idea; I have added a table (line 625)
On page 5, line 254: “This does not only depend only on social and medical norms, but also on moral convictions of what good care entails.” The author should delete the latter "only".
Done!
The author should double check the format of the references.
I have checked and revised them.
Regards
Reviewer 3 Report
Thank you for the opportunity to read and review this very interesting paper. I loved many parts of it, especially the interesting data and analyses.
I think the core argument (about justification work) is quite strong and persuasive, but there must be more work done to shape how you present it, in order to strengthen the paper itself as a whole, not just the core findings. Mainly, this has to do with presenting a more nuanced understanding of the field of healthcare improvement (and the triple aim approach) that you set your approach against.
In the following, I will paste the notes I made as I read through the paper, including short comments and corrections, and at the end I have put in a more extensive and detailed comment.
-
[Line 46-47] This sentence does not seem to follow from the previous point. Do you mean to say instead that to prevent depletion is thus rational?
-
[Lines 73-76] An important point indeed!
-
[line 95] – what is a ‘low complex problem’?
-
Line 97 – typo: quality of care
-
Line 99-100 – You might want to refine what you say here – your approach is not in contrast to ‘the Triple Aim approach’, but rather is in contrast to the ‘outcome measures-focused’ approach. The triple aim is very closely associated with quantitative data, especially regarding their view of ‘a learning system’ as you note, but this is not their central argument – which is more about balancing a number of priorities, including costs. It may be more useful/defensible to challenge the ‘outcomes measures’ part without also challenging the ‘triple aim’ itself. See big comment at the end.
-
108 – Typo: physiotherapists
-
134 – What were the outcome measures used to evaluate this intervention? If you can outline how this was intended to be assessed, then you can more effectively contrast/compare what you were able to do, using exnovation.
-
156 – Typo: patients and professionals
-
157-159 – Sentence is unclear. Please rephrase, check grammar
-
165-166 – As a reader I feel like I would like more of an explanation here about the terms ‘resistance and accommodation’ – with the understanding that this might be explored later in the discussion. Maybe a sentence or two?
-
183 – Typo: spent
-
192-193 – were the pre and post-observation interviews held with both practitioner and patient together or separately?
-
194-107 – Great point about usefulness of role as outsider!
-
246 – This line looks incomplete, difficult to read, please check phrasing. Who refers to what? Also missing a quotation mark at the start of the quote.
-
259 – I love the notion of ‘durable decisions’
-
263-275 – I like what you say here, but your references are incorrect. Bardram and Bossen paper is listed to support Boltanski and Thévenot’s work. Please correct.
-
285 – why only those 3 ‘orders of worth’? Were those the only ones found, or were only these 3 chosen for the analysis for a particular reason? Please clarify either way. Also, could be useful to comment on how these 3 orders of worth ‘fit’ conceptually in relation to the triple aim (cost, care, health)
-
304 – As a reader I would appreciate a brief paragraph at the start, to prepare me for what is being presented in this results section – e.g. what are the headings? Themes? What data will I be shown (field notes? Vignettes? Quotes etc). Likewise at the start of each ‘section’, a sentence or two to orient the reader about what this part will be about, rather than going straight to the data.
-
408 – Typo: this
-
405 – did the civic order of worth emerge, or was it co-created by the ‘mangle’ of her previous experience, the actions of the rheumatologist, the presence of a researcher asking questions, and so on? Also, could there be multiple orders of worth in play?
-
467 – The sentence refers to section 3.1 – I was able to find the section that you meant, but for a moment I was confused – the observations are labelled in a similar way to the ‘sections’, and for the reader, this labelling is more apparent for the observations not the sections. Consider relabelling the observations – 1, 2 and 3, or A B and C?
-
468-470; 528-530 – It seems to me that the doctor and patient are also enrolling the materiality of the patients’ bathroom tiles into the mangle, not just navigating the absence of the high-tech equipment.
-
496-510 – I wonder why this example is not also a matter of the order of efficiency (as well as emotional proximity)? Are there also elements of equality involved – in that the same patient would have received better(?) care at the specialist eye clinic? Also, the explanation of caring for the patient ‘as if he is a child’ sounds a bit strange – is there a reason to describe it that way?
-
541-546 – This strikes me as similar to the patient in the previous example who may have likewise been reflecting on her consult in light of her previous experience, perhaps because a researcher asked? This reminds me of the role of sense-making, and accountability (e.g. to a researcher) that creates the requirements for justification (outside of the work done within the consultation).
It also strikes me that the discussion about the second example/observation is relevant to the literature on a quadruple aim that includes health professionals’ satisfaction
-
556 – So interesting. Here we have orders of worth in conflict.
-
564 – rephrase sentence for grammar – move the word ‘also’
-
622 – Typo? Bowel instead of bowl?
-
707-718 – I thought this was very well-written and convincing.
-
724-735 – This was less convincing. I think you need to do some more ‘unpacking’ of the project of healthcare improvement (and innovation) to be able to situate your very interesting and useful work more appropriately in this area.
For instance, the IHI (organisation that created the triple aim) has its origins in the continuous quality improvement (QI) work of industrialists like Shewhart and Deming in the 1930s and 1980s, who were very interested in identifying and reducing unwarranted variation. This explains (in part) its push for the ‘science of improvement’ by taking measurements and conducting iterative experiments (PDSA cycles). The triple aim itself was created in this organisation in contrast to a previous set of aims for healthcare improvement (that focused on aspects of healthcare quality alone). So, the introduction of cost and population health was the innovation here. The reliance on quantitative measurement is part of the broader project of evaluating healthcare improvement by numbers. This goes back to the history of quality improvement, and is bigger than the triple aim itself. It also links to the ‘evidence-based practice’ movement, and the hierarchy of evidence (with the RCT on top).
Although the triple aim is certainly very popular and perhaps the latest ‘face’ of this ‘science of QI’ movement, I think it’s important not to confuse it with its longer and broader history. One aspect of the triple aim approach that might be more useful to focus on, maybe, is the concept of a ‘learning system’ as mentioned in that paper by Whittington et al (2015). This links with the very popular ‘Learning Health System’ project coming out of the UK (https://learninghealthcareproject.org/background/learning-healthcare-system/).
Also, by this point, I am left uncertain as to the point of describing and framing this work with the tragedy of the commons. The triple aim attempts to address the tragedy of the commons by making the sustainability of healthcare interventions rational to decision-makers. You have very nicely described how what is ‘rational’ to the actors within the intervention is unpredictable and emergent. However, the comparisons or links are not made in this paper between the concepts of ‘health, care, and cost’ as (triple aim) justifications, and the ones you identify (equality, emotion/trust and efficiency). I’m not sure if this suggestion suits what you are trying to do, but it seems a wasted opportunity, especially when a stated key challenge of the triple aim is reaching a balance between the three, and what you describe here is the tricky work of balancing (sometimes multiple) orders of worth.
Finally, I think there is also a useful point to be made about the role of accountability, as I briefly noted above. Consider why outcomes measures are so valued and pervasive – they fit very well the needs of ‘centres of calculation’ (see Latour’s work on immutable mobiles), and have become the way in which health services and practitioners account for the quality of their care (and cost). Consider also the role that the researcher played here, as someone who was asking participants to give accounts of their behaviours and thoughts about the consultations. It could be very useful to consider how exnovation is a process of accountability that serves purposes other than calculation (you discuss some of this already), and to consider how it serves the broader aim of understanding the sustainability of innovations. You may not agree and that is fine, this is not a required change, but I think it would be very useful to at least consider how this corresponds with what you are trying to say.
-
Finally, reference #15 is missing an author, please fix that.
Author Response
Dear reviewer,
I was really moved to see how much time and effort you have spent to give such great, constructive and insightful feedback to the manuscript. Such great suggestions. I have incorporated most of them. I think that the paper has really improved. I hope you agree and that I have addressed the issues sufficiently.
In the below I explain for each of your comments (cursive) what I have done to address them, except for the typos and compliments :).
[Line 46-47] This sentence does not seem to follow from the previous point. Do you mean to say instead that to prevent depletion is thus rational?
I have rephrased (and shortened) the paragraphs. I hope it is clearer now. (lines 28-40)
[line 95] – what is a ‘low complex problem’?
Rephrased (lines 464-465)
Line 99-100 – You might want to refine what you say here – your approach is not in contrast to ‘the Triple Aim approach’, but rather is in contrast to the ‘outcome measures-focused’ approach. The triple aim is very closely associated with quantitative data, especially regarding their view of ‘a learning system’ as you note, but this is not their central argument – which is more about balancing a number of priorities, including costs. It may be more useful/defensible to challenge the ‘outcomes measures’ part without also challenging the ‘triple aim’ itself. See big comment at the end.
I have now focused more on the idea of the learning system and the outcome measures. I hope it is more convincing now (lines 118-149.
134 – What were the outcome measures used to evaluate this intervention? If you can outline how this was intended to be assessed, then you can more effectively contrast/compare what you were able to do, using exnovation.
I have added some more information (462-464). However, I feel outlining how this was done and effectively comparing it with the exnovation method is beyond the scope of this article because my aim is to show in more general the flaws of the triple aim approach, not of the specific way in which the team that monitored the Primary Care Plus centers has done it.
165-166 – As a reader I feel like I would like more of an explanation here about the terms ‘resistance and accommodation’ – with the understanding that this might be explored later in the discussion. Maybe a sentence or two?
Added (612-614)
192-193 – were the pre and post-observation interviews held with bothpractitioner and patient together or separately?
Added (line 618)
285 – why only those 3 ‘orders of worth’? Were those the only ones found, or were only these 3 chosen for the analysis for a particular reason? Please clarify either way. Also, could be useful to comment on how these 3 orders of worth ‘fit’ conceptually in relation to the triple aim (cost, care, health)
Added (728-729)
304 – As a reader I would appreciate a brief paragraph at the start, to prepare me for what is being presented in this results section – e.g. what are the headings? Themes? What data will I be shown (field notes? Vignettes? Quotes etc). Likewise at the start of each ‘section’, a sentence or two to orient the reader about what this part will be about, rather than going straight to the data.
Added (lines 801-805)
405 – did the civic order of worth emerge, or was it co-created by the ‘mangle’ of her previous experience, the actions of the rheumatologist, the presence of a researcher asking questions, and so on? Also, could there be multiple orders of worth in play?
Nice way of phrasing it. I have added the suggestions (lines 934-937).
467 – The sentence refers to section 3.1 – I was able to find the section that you meant, but for a moment I was confused – the observations are labelled in a similar way to the ‘sections’, and for the reader, this labelling is more apparent for the observations not the sections. Consider relabelling the observations – 1, 2 and 3, or A B and C?
I also think it is a bit confusing, however, I have tried to follow the journal’s template as closely as possible. I have added tree diagrams now – a suggestion by another reviewer, I hope this makes it a bit easier to read.
468-470; 528-530 – It seems to me that the doctor and patient are also enrolling the materiality of the patients’ bathroom tiles into the mangle, not just navigating the absence of the high-tech equipment.
Great suggestion! (1005-1012)
496-510 – I wonder why this example is not also a matter of the order of efficiency (as well as emotional proximity)? Are there also elements of equalityinvolved – in that the same patient would have received better(?) care at the specialist eye clinic?
I decided to focus on the domestic order, because it seems the dominant one here.
Also, the explanation of caring for the patient ‘as if he is a child’ sounds a bit strange – is there a reason to describe it that way?
That is the specific way that Boltanski and Thevenot explain the domestic order. However, I understand it is a bit strange, and I have changed it in the revised version (1038-1039)
541-546 – This strikes me as similar to the patient in the previous example who may have likewise been reflecting on her consult in light of her previous experience, perhaps because a researcher asked? This reminds me of the role of sense-making, and accountability (e.g. to a researcher) that creates the requirements for justification (outside of the work done within the consultation)
Good point! There are some small comments about this - and a remark in the discussion (1369-1370)- but I have not further added it to the text because of the limited space of this paper. But I do think the reflective conversations were part of the justification work and by itself helped to improve the practices.
It also strikes me that the discussion about the second example/observation is relevant to the literature on a quadruple aim that includes health professionals’ satisfaction
This is an interesting and relevant point, but unfortunately I think that there is no space to address it in this paper.
724-735 – This was less convincing. I think you need to do some more ‘unpacking’ of the project of healthcare improvement (and innovation) to be able to situate your very interesting and useful work more appropriately in this area.
I have revised the discussion and conclusion sections. I hope it is sufficiently convincing now. (lines 1368-1460)
For instance, the IHI (organisation that created the triple aim) has its origins in the continuous quality improvement (QI) work of industrialists like Shewhart and Deming in the 1930s and 1980s, who were very interested in identifying and reducing unwarranted variation. This explains (in part) its push for the ‘science of improvement’ by taking measurements and conducting iterative experiments (PDSA cycles). The triple aim itself was created in this organisation in contrast to a previous set of aims for healthcare improvement (that focused on aspects of healthcare quality alone). So, the introduction of cost and population health was the innovation here. The reliance on quantitative measurement is part of the broader project of evaluating healthcare improvement by numbers. This goes back to the history of quality improvement, and is bigger than the triple aim itself. It also links to the ‘evidence-based practice’ movement, and the hierarchy of evidence (with the RCT on top).
Although the triple aim is certainly very popular and perhaps the latest ‘face’ of this ‘science of QI’ movement, I think it’s important not to confuse it with its longer and broader history. One aspect of the triple aim approach that might be more useful to focus on, maybe, is the concept of a ‘learning system’ as mentioned in that paper by Whittington et al (2015). This links with the very popular ‘Learning Health System’ project coming out of the UK (https://learninghealthcareproject.org/background/learning-healthcare-system/).
Also, by this point, I am left uncertain as to the point of describing and framing this work with the tragedy of the commons. The triple aim attempts to address the tragedy of the commons by making the sustainability of healthcare interventions rational to decision-makers. You have very nicely described how what is ‘rational’ to the actors within the intervention is unpredictable and emergent. However, the comparisons or links are not made in this paper between the concepts of ‘health, care, and cost’ as (triple aim) justifications, and the ones you identify (equality, emotion/trust and efficiency). I’m not sure if this suggestion suits what you are trying to do, but it seems a wasted opportunity, especially when a stated key challenge of the triple aim is reaching a balance between the three, and what you describe here is the tricky work of balancing (sometimes multiple) orders of worth.
Finally, I think there is also a useful point to be made about the role of accountability, as I briefly noted above. Consider why outcomes measures are so valued and pervasive – they fit very well the needs of ‘centres of calculation’ (see Latour’s work on immutable mobiles), and have become the way in which health services and practitioners account for the quality of their care (and cost). Consider also the role that the researcher played here, as someone who was asking participants to give accounts of their behaviours and thoughts about the consultations. It could be very useful to consider how exnovation is a process of accountability that serves purposes other than calculation (you discuss some of this already), and to consider how it serves the broader aim of understanding the sustainability of innovations. You may not agree and that is fine, this is not a required change, but I think it would be very useful to at least consider how this corresponds with what you are trying to say.
These are all very interesting and relevant points. I have added as much as I could to the discussion of the paper (1296-1326).
Finally, reference #15 is missing an author, please fix that.
Apologies!!!!!!
Round 2
Reviewer 1 Report
Authors made almost all required revisions and responded to all reviewers' comments and questions. it has been highly increased.
This manuscript is a resubmission of an earlier submission. The following is a list of the peer review reports and author responses from that submission.
Round 1
Reviewer 1 Report
Congratulations! It is very well writen article, interesting and innovative. It focus and reflects on very important issue sustanable care. I hope the results will be inspiring and can help to improve health care.
Reviewer 2 Report
Review of the article: Overcoming the Tragedy of the Health Care Commons by Experimentation. How Exnovations Brings Unexpected Problems and Solutions into View
I have read and examined very carefully the work received and unfortunately my opinion about it is not favourable.
The manuscript suffers from a number of very important flaws.
First, the paper is poorly written. The English language is not correct, although this can be corrected through an editing process.
However, the article has another problem in its wording, which would require a new elaboration. I am referring to the fact that the article is not written according to scientific language, but in common language. For example, in a scientific text, the first person singular should never be used.
The author also does not use inclusive language, as required by today's scientific journals. For example, page 4, line 152.
The manuscript uses acronyms that are not known to all readers. For example, RA (page 6, line 291) and MRI (page 6, line 299).
Bibliographic references are not correctly cited. For example, page 2, lines 45 and 95: “Berwick et al it is possible…” and “Iedema et al argue these…”.
The “Triple Aim” is not defined until page 2, line 63. However, it is used previously and the reader unknown its meaning.
The article lacks a literature review section following the Introduction. This implies that there are no hypotheses to be contrasted.
No scientific methodology was evident in the preparation of this article. If there are interviews, a questionnaire should be defined. The data would have to be tabulated, some reliable statistical treatment would have to be carried out, etc. None of this has been done.
In short, I consider that this manuscript does not correspond to a scientific paper.
I hope that my comments will be understood by the author in a positive sense, that he will use them to remodel the work and to produce a manuscript that can be publishable.
Reviewer 3 Report
Dear Authors,
Contribution entitled:
"Overcoming the Tragedy of the Health Care Commons by Ex-2 perimentation. How Exnovation Brings Unexpected Problems 3 and Solutions into View"
is indeed an intriguing one depicting some sound methodological arbitrations on cost of health care and innovations in medicine.
I believe it could fill certain knowledge gap.
Its one core weakness is the evidence base cited which should be substantially diversified and expanded.
It leans too heavily to the academic sources based in rich OECD nations while far more representation of seminal literature outsourcing from LMICS and Emerging Makrets in aprticular should be introduced.
Thus an article would gain additional methodological reliability and comprehensiveness.
For this purpose I warmly recommend consideration for inclusion of at least several sources listed below alongside few additional ones at authors own disposal:
https://agsjournals.onlinelibrary.wiley.com/doi/abs/10.1111/jgs.15591
https://www.jonskinner.org/s/bynum_et_al-2018-journal_of_the_american_geriatrics_society.pdf
https://www.tandfonline.com/doi/full/10.1080/13696998.2019.1600523
https://www.bmj.com/content/359/bmj.j4695.abstract
https://resource-allocation.biomedcentral.com/articles/10.1186/s12962-020-00210-2
https://www.taylorfrancis.com/chapters/edit/10.1201/9781315565200-5/resources-strength-exnovation-hidden-competences-preserve-patient-safety-jessica-mesman
https://www.frontiersin.org/articles/10.3389/fpubh.2016.00115/full
https://link.springer.com/article/10.1007/s11625-019-00681-0
https://onlinelibrary.wiley.com/doi/abs/10.1111/1468-0009.12213
Reviewer 4 Report
I would like to thank the author of this article for the entertaining description of their work. I read it as a novel or a biographical work and I do question the scientific method used hence the validity of the study. It will be a trap to discuss the findings and trying to explain them but question the significance.
I may emphasize the simple fact that (health) and (medicine) are the same but two different issues. Tragedy of the commons definition should be clarified in the economical context and applied to the medical services.
With missing dates of data collection I wonder how significant is the study after 2020? The researcher describes three cases reported and exhibited reflections and explanations of different attitudes. The study is missing few items: Study Aim, Conclusion and Recommendations. Please find attached my comments. A social mobility reflection actually updated since the erupted coronavirus crisis measures may give the article some insight.
Best regards

Round 2
Reviewer 2 Report
Dear Author,
I understand that my opinion may be disappointing to you, but I have to be honest in my approach.
I reiterate my previous assessment and, even acknowledging your effort, there are important deficiencies that should be remedied to consider the article publishable.
I regret, therefore, that I cannot give a favorable opinion.
In any case, it is the Editor's decision to accept or reject publication.
Best regards
Reviewer 4 Report
Dear author, This is my second review of your article : Overcoming the Tragedy of the Health Care Commons by Experimentation. Bringing Unexpected Problems and Solutions into View. As previously mentioned the title do not correspond to the study investigation and design: The tragedy of the health care commons cannot be explained or solved by solely analyzing, observing or reflecting over physicians behavioral patterns and patients attitudes at some statistically not representative sample.
Line 102: 2. Aim of the Study: By tracking, analyzing and reflecting on the mundane habitual behaviors of patients and professionals in three Primary Care Plus consultations in detail, I aim to bring into view what is rendered invisible with the Triple Aim approach. As such, I aim to contribute to a broad perspective on sustainability in health care. (The study aim is ill defined and unmet)
Line 107: 3. Materials and Methods: 3.1: Study design: what you named as ethnographic case study may be used in the introduction section but what i read is not a study design. Consider sampling methodology, sample size and description instead. Data collected details instead of exnovation methodology. This section is not clear: It may gain significance by demonstrating the kind of reflections different attitudes and behavioral patterns. The dissertation about 3.4: data analysis and 3.5: analytical lens may be more explicit about the kind of data instead. 4.Results: are not relevant, 4.1 crafting durable decisions , 4.2 unforeseen problems and, 4.3 creating new care routines are not results of the study but three cases reported. your observations and descriptions are enjoyable but no scientific conclusion can be based on such approach. we expect to see tables and statistical analysis of grouped observations of a relevant sampling methodology. 5. Discussion of the actual finding of a research study is based on the discussion of the findings and compare them with similar studies. You might have explored in details three consultations through what you called your analytical lens and we can discuss every quote and observation you described but they remain limited in the broader concept in health management and administration. Your observations and adventure may be very helpful as pilot study for a well configured actions like a representative sample of your subjects of study (physicians and patients) with well defined questionnaires and valid data. Separate items as 5.conclusion and 6.recommendations may help in focusing on the findings of the actual study. Again, some minor issues mentioned in the first review like the page number of the references to be moved to the reference endnote.
I think that your article needs extensive efforts to be ready for publishing.
Best Regards